# The Importance of Accurate Early Diagnosis and Eradication in *Helicobacter pylori* Infection: Pictorial Summary Review in Children and Adults

**DOI:** 10.3390/antibiotics12010060

**Published:** 2022-12-29

**Authors:** Cristina Maria Marginean, Ramona Cioboata, Mihai Olteanu, Corina Maria Vasile, Mihaela Popescu, Alin Iulian Silviu Popescu, Simona Bondari, Denisa Pirscoveanu, Iulia Cristina Marginean, George Alexandru Iacob, Mihai Daniel Popescu, Mihaela Stanciu, Paul Mitrut

**Affiliations:** 1Department of Internal Medicine, University of Medicine and Pharmacy of Craiova, 200349 Craiova, Romania; 2Department of Pneumology, University of Pharmacy and Medicine Craiova, 200349 Craiova, Romania; 3Department of Pediatric and Adult Congenital Cardiology, Bordeaux University Hospital, 33600 Pessac, France; 4Department of Endocrinology, University of Medicine and Pharmacy of Craiova, 200349 Craiova, Romania; 5Department of Radiology, University of Medicine and Pharmacy of Craiova, 200349 Craiova, Romania; 6Department of Neurology, University of Medicine and Pharmacy of Craiova, 200349 Craiova, Romania; 7Faculty of Medicine, University of Medicine and Pharmacy of Craiova, 200349 Craiova, Romania; 8Department of Endocrinology, Faculty of Medicine, Lucian Blaga University of Sibiu, 550169 Sibiu, Romania

**Keywords:** *H. pylori*, children, diagnostics test, chronic gastritis, antibiotics therapy

## Abstract

Among the most widespread childhood infections, *Helicobacter pylori* (*H. pylori*) develops potentially life-threatening conditions in adults if not appropriately treated. Helicobacter pylori is a common human pathogen that was first described in the stomach many years ago. The discovery of *H. pylori* was crucial in gastroenterology; this bacterium is associated with chronic gastritis, peptic ulcers, gastric cancer, and lymphoid tissue lymphoma related to the gastric mucosa. Studies published so far estimate that approximately 10% of subjects infected with *H. pylori* develop a peptic ulcer, and 1–3% of subjects develop gastric cancer. The clinical manifestations are variable and characteristically depend on the individual factors of the host. Various methods of detection and diagnosis of *H. pylori* infection have been developed, each with advantages, disadvantages, and/or limitations. Available diagnostic tests are usually performed using invasive (endoscopy, biopsy, rapid urease test, cultures, and molecular tests) and noninvasive methods (urea breath test, stool antigen examination, and serological and molecular tests). Although there is extensive accessibility for diagnosing and treating *H. pylori* infection, the prevalence of antibiotic resistance is not negligible. Thus, numerous studies and meta-analyses are focused on a new orientation of gastroenterologists in diagnosing and treating *H. pylori* infections. A fascinating perspective hypothesis is the administration of probiotics to reduce *H. pylori* adhesion to gastric epithelial cells, preventing *H. pylori* colonization, especially in children, or reinfection with *H. pylori* in high-risk adult patients.

## 1. Introduction

Among the most widespread childhood infections, *Helicobacter pylori (H. pylori*) develops potentially life-threatening conditions in adults if not appropriately treated. *H. pylori* is a common human pathogen that was first described in the stomach many years ago [1]. In 1983, *H. pylori* was isolated from the gastric antrum by Marshall and Warren and described as a Gram-negative, flagellated, and microaerophilic bacterium [2]. The discovery of *H. pylori* is of crucial importance in gastroenterology; this bacterium is associated with chronic gastritis, peptic ulcers, gastric cancer, and lymphoid tissue lymphoma associated with the gastric mucosa [3].

This paper aims to summarize the most recent findings in the treatment and diagnosis of *H. pylori* infections in adult and pediatric populations. This narrative review presents data on prevalence, clinical manifestations, diagnostic methods, and the most recent treatment regimens. Through this paper, we aim to fix the gaps in the current literature and raise the alarm among adult and pediatric patients who require correct treatment in order not to develop complications in adult life.

It is essential to understand that antibiotic regimens have increased antibiotic resistance tremendously, and therefore, often, the proper treatment for *H. pylori* infection can be a challenge for the doctor.

## 2. Prevalence and Mode of Transmission

Various studies over time have shown that the prevalence of *H. pylori* infection varies according to age, geographic area, and socioeconomic status [4]. 

Population studies suggest that at least 50% of people are infected with *H. pylori*.

Oral–oral is the main route of transmission of *H. pylori*, explaining the incidence of infection among members of the same family [5]. Fecal–oral transmission is the second route of infection described after exposure to contaminated water, mainly due to poor sanitation conditions [6]. 

For this reason, it is essential to evidentiate that improving socioeconomic status and living conditions dramatically influence the reduction of the prevalence of *H. pylori* infection [7]. 

## 3. Pathogenesis

*H. pylori* is a mobile, microaerophilic, Gram-negative, rod-shaped bacterium capable of colonizing the acid environment existing in the stomach. In most cases, *H. pylori* infection is contracted in childhood. It is estimated that 30% of the young population is or will become infected with *H. pylori* at some point during their lifetime. The interactions between the pathogen agent’s virulence factors and the host’s immune response seem crucial in developing the pathologies associated with this infection, such as gastritis, peptic ulcers, or gastric cancer (Figure 1). Studies published so far estimate that approximately 10% of subjects infected with *H. pylori* develop peptic ulcers, and 1%-3% of subjects develop gastric cancer [8,9].

Inside the stomach, *H. pylori* produces urease, which favors the hydrolysis of urea with the generation of carbon dioxide and ammonia. Ammonia neutralizes gastric acid, and carbon dioxide prevents bacteria’s survival in the stomach by raising the acidity of the medium. *H. pylori*, due to its helical shape and flagellar movements, penetrates the mucus layer and gastric epithelium, and secondarily, the epithelial cells, determined by a synergic action of adhesins and membrane proteins, subsequently triggering an inflammatory response in the host cells. Gastric intestinal metaplasia, peptic ulcers, chronic atrophic gastritis, gastric cancer, and lymphoid tissue lymphomas (MALT) [10] were complications that led the World Health Organization (WHO) to classify *H. pylori* as one of the carcinogenic agents of class 1 [11]. Previous studies have concentrated mainly on the role of *H. pylori* in developing inflammations and gastric cancer and demonstrated that eradicating *H. pylori* can reduce the incidence of gastric cancer [12] and atrophic gastritis [13]. The guidelines regarding the management of precancerous epithelial diseases (MAPS II) in 2019 [14] recommended, as primary objectives, the prevention of *H. pylori* infection, considering its proven role in the process of gastric carcinogenesis, as well as the eradication of *H. pylori* in patients with known gastric or intestinal metaplasia and chronic atrophic gastritis.

## 4. Clinical Manifestations and Diagnosis

According to the pathogenic mechanisms, *H. pylori*-positive subjects can present various clinical manifestations [15]. The clinical manifestations are variable and characteristically depend on the individual factors of the host, as the type of initial damage to the gastric mucosa is correlated with the progression of various gastric diseases. The predominant development of antral gastritis is associated with the appearance of a peptic ulcer. In contrast, corpus gastritis and multifocal atrophy correlate with gastric ulcers, gastric atrophy, intestinal metaplasia, and gastric carcinomas [16] (Figure 2). The most frequent clinical manifestations in medical practice are chronic gastritis and peptic ulcers; for this reason, detecting and eradicating the infection are essential steps in managing these conditions [17]. 

Recent studies have associated *H. pylori* infection with a comprehensive pathology of extra digestive effects, with the condition being related to allergic, neurological, dermatological, ocular, hematological, cardiovascular, and metabolic diseases, a fact that amplified the detection and eradication importance [18].

## 5. Diagnostic Methods in *H. pylori* Infection

Various methods of detecting *H. pylori* infection have been developed, each with advantages, disadvantages, and/or limitations. The first choice depends both on the availability and accessibility of the laboratories and the clinical conditions of the patients. Additionally, available diagnostic tests that are usually performed include invasive (endoscopic) and noninvasive methods. Non-invasive diagnostic tests are represented by the urea breath test, stool antigen examination, and serological and molecular tests. Invasive tests include an endoscopy, a biopsy, a rapid urease test, cultures, and molecular tests [19]. 

### 5.1. Noninvasive Diagnostic Tests

#### 5.1.1. Urea Breath Test

The urea breath test (UBT) was the first method of diagnosis and still represents the most popular noninvasive test for diagnosing *H. pylori* infection. UBT presents multiple advantages in detecting *H. pylori* infection, especially in children and teenagers, being a simple, non-invasive, and reliable method, with a sensitivity and specificity of 75% to 100% [20]. 

A recently published meta-analysis evaluating the diagnostic accuracy of *H. pylori* infection using UBT in adult patients with specific dyspeptic symptoms revealed a sensitivity of 96% and a common specificity of 93% [21]. 

UBT is also helpful for epidemiological studies and for evaluating eradication therapy efficiency [16].

#### 5.1.2. Stool Antigen Test

The stool antigen test (SAT) is a noninvasive method with remarkable sensitivity and specificity, 94% to 97%, used to diagnose *H. pylori* infection [22]. 

Also used to confirm eradication after anti-infective therapy, the cumulative sensitivity and specificity for the monoclonal SAT were 93% and 96%, respectively, according to data published in global meta-analyses [22]. 

The accuracy of the stool antigen diagnosis is influenced by factors such as the use of antibiotics, PPI, or other drugs such as N-acetylcysteine, and the coexistence of complications such as upper digestive hemorrhage. Before testing, sample preservation, temperature, and transport time can influence the diagnostic accuracy of the SAT [23,24]. 

#### 5.1.3. Serological Tests

Currently, numerous serological tests are available based on the detection of anti-*H. pylori* IgG antibodies; the EIA test is the most used. Serological tests are frequently used in the screening of epidemiological studies due to their acceptability for patients, prompt result, and inexpensive cost.

The accuracy of serological tests is not affected by digestive hemorrhaging, gastric atrophy, antibiotics, or PPIs, which cause false negative results in other methods.

The disadvantage of serological tests is the impossibility of evaluating the efficacy of the eradication therapy because the levels of anti-*H. pylori* antibodies can persist in the blood for a long time, even after the eradication of the infection. Antibody-based tests do not distinguish between active infection and previous exposure to *H. pylori*, so further confirmation is required before the initiation of eradication therapy [16]. 

The determination of *H. pylori* antibodies has a decisive role in studies related to the pathogenesis of the infection and the assessment of virulence factors because specific antigenic proteins can be detected via immunological techniques, thus conferring an additional diagnostic value. Potential biomarkers have been tested to identify subjects infected with *H. pylori* strains with a high risk of developing complications and assess the prognosis of diseases associated with *H. pylori* infection.

Thereby, in patients affected by *H. pylori*, the detection of serum CagA, VacA, and GroEL antibodies was correlated with the development of gastric precancerous lesions and even gastric cancer, making these serum markers potential predictive factors for patients infected with high-risk strains [25,26]. 

#### 5.1.4. Polymerase Chain Reaction

A PCR test detects *H pylori* in stool and is a reliable and fast technique that offers the advantage of identifying specific genotypes, with a possible role in determining the antibiotic resistance of bacteria [27,28].

The detection of infection through PCR in the oral cavity uses *H. pylori*-specific primer sets based on specific complete genome sequences of 48 *H. pylori* strains, increasing the diagnostic accuracy [29].

### 5.2. Invasive Tests

#### 5.2.1. Endoscopy

Conventional endoscopic examination is routinely performed to evaluate lesions associated with *H. pylori* (chronic atrophic gastritis, peptic ulcers, MALT lymphomas, and gastric cancer). Endoscopy is also used for taking biopsy samples from the gastric mucosa for further determinations, such as rapid urease tests, histopathological examinations, cultures, and molecular methods. The biopsy is preferably performed from the gastric antrum.

Chromoendoscopy with phenol red is a method used for detecting *H. pylori* infection due to the specific urease activity of the bacterium, but its sensitivity (73–81%) and specificity (76–81%) are low [30,31].

Confocal laser endomicroscopy (CLE) represents an endoscopic method that allows the histological examination of the gastric mucosa. Characteristic for the positive diagnosis of *H. pylori* infection is the presence, at laser confocal endomicroscopy, of white spots, neutrophils, and sometimes micro-abscesses. The accuracy, sensitivity, and specificity of this method have been evaluated in various studies at 92.8%, 89.2%, and 95.7%, respectively [32].

#### 5.2.2. Histological Examination

The histological examination is the gold standard for diagnosing *H. pylori* infection. The examiner’s experience influences the accuracy of the diagnosis, size, site, number of biopsies, and staining procedures, as well as the use of inhibitors of the proton pump (PPI) and antibiotics.

#### 5.2.3. The Rapid Test with Urease

The rapid urease test (RUT) is considered the most useful invasive test in the diagnosis of *H. pylori* infection because it is quickly available, not expensive, easy to perform, and, in particular, has a sensitivity of over 85–95% and a specificity between 95% and 100%, with increased accuracy with the number of gastric biopsies [33].

#### 5.2.4. Microbiological Culture

The cultivation of *H. pylori* from the gastric biopsy sample is a diagnostic method with a specificity of approximately 100% but less sensitivity, with variability between 85% and 95%. Culturing allows the isolation of *H. pylori* for subsequent phenotypic and genotypic characterization, with a role in the efficiency of the therapeutic scheme. The cultivation of *H. pylori* provides the ability to manage and assess antibiotic resistance in extensive observational studies [16].

#### 5.2.5. Polymerase Chain Reaction

PCR is widely used to detect *H. pylori* infection from gastric biopsy samples, saliva, and stool. Compared to conventional tests, PCR offers a sensitivity and specificity above 95%. PCR provides information on *H. pylori’s* virulence factors by detecting specific mutations that determine antibiotic resistance, such as resistance to macrolides and fluoroquinolones [34,35].

## 6. Therapeutic Management

Currently, it is considered appropriate to detect and treat the infection in the following categories of patients:

Patients diagnosed with intestinal metaplasia, MALT lymphoma, gastric atrophy, or early gastric cancer; elderly patients with dyspeptic syndrome or with suspicious lesions described at endoscopy; young symptomatic patients diagnosed through non-invasive tests; chronic users of NSAIDs or anticoagulants, especially those with a history of gastritis or peptic ulcers; patients diagnosed with iron deficiency anemia, a deficit of the B12 vitamin, or idiopathic thrombocytopenic purpura.

According to recent studies, a comprehensive approach toward *H. pylori* testing and treatment policies is recommended, including tests for family members of infected patients and high-risk populations (i.e., immigrants from countries with a high prevalence of infection), especially in countries where there is an increased rate of gastric cancer and where screening programs have been developed for the entire population. The ideal methods of diagnosis, the need for cultures and an antibiogram before prescribing the treatment, and the confirmation of eradication still need to be elucidated, requiring extensive additional studies.

Most therapeutic regimens are recommended empirically, with no data on the resistance to specific classes of antibiotics. According to the latest recommendations, the optimal treatment for *H. pylori* infection might have an eradication rate of at least 90% [36].

The published data describe different concepts of medical practices in different regions of Europe, with consequent heterogeneous results: seven-day regimens are very common in South-East Europe (60%) and less frequently used in South-West Europe (1.7%). An observation was made regarding triple therapy: it is very rarely prescribed in Central and South-West Europe and much more frequently in East, South-East, and North Europe.

*H. pylori* treatment uses a combination of antimicrobial and antisecretory agents (Figure 3). To achieve the bactericidal effect of antimicrobial agents, it is necessary to increase the gastric pH, which is provided by administering an antisecretory treatment. Probiotics are used as adjunctive therapy to eradicate *H. pylori*, although their pathogenic mechanism is unclear [37].

As resistance against single antibiotic therapy develops rapidly, a combination of several antibiotics is recommended [38].

Triple therapy, including clarithromycin and amoxicillin, was the most frequently used regimen, with eradication rates below 86%. Among all antibiotic regimens prescribed, only concomitant therapy with clarithromycin, amoxicillin, and metronidazole achieved a 90% eradication rate. In regions where bismuth salt was used in quadruple therapy, eradication rates were highest, exceeding 90%.

Thus, according to the most recent published data [37], quadruple therapy is recommended, being the only one that achieves an eradication rate of approximately 90%:

Quadruple treatment with PPIs, amoxicillin, clarithromycin, and metronidazole, for 14 days.

Triple therapy plus bismuth salt: PPI, bismuth salt, amoxicillin, and clarithromycin, for 14 days.

Bismuth quadruple therapy: PPI, bismuth salt, tetracycline, and metronidazole, for 10 days.

If the first line of treatment fails, the Maastricht Consensus V recommends rescue therapy with PPIs, amoxicillin plus quinolone (levofloxacin or moxifloxacin), for 10–14 days.

If the second line of treatment also fails, cultures with antibiotic susceptibility tests or molecular determinations are recommended [16].

### 6.1. Antibiotic Resistance of H. pylori

Currently, drug resistance of *H. pylori* is associated with antibiotic resistance included in eradication regimens (clarithromycin, metronidazole, amoxicillin, fluoroquinolones, and tetracycline). The mechanism of antibiotic resistance refers to structural changes in the gene sequence. Intracellular drug activation is inhibited, with subsequent enzymatic deactivation.

The efficacy of antibiotic administration is reduced in the acidic environment of the stomach, a fact also explained by the thickness of the gastric mucosa of approximately 200 μm, which leads to a decrease in the eradication rate.

This describes the capacity of *H. pylori* to form biofilms composed of bacteria and an extracellular matrix, self-secreted, that adhere to various surfaces. Biofilms increase the antibiotic resistance of *H. pylori* by creating a barrier that reduces the penetration of the antibiotic, the promotion of gene mutation, and the overexpression of the efflux pumps involved in drug resistance. The extracellular matrix also creates an effective but non-specific barrier that does not allow antibiotic penetration, favoring efflux pumps’ overexpression. This encapsulation of *H. pylori* causes a significant increase in its viability. Considering these particular aspects of *H. pylori’s* ability to develop microfilms that can increase antibiotic resistance, it becomes imperative to investigate alternatives to improve the efficiency of drug administration, annihilate virulence factors, and implicitly contribute to eradicating the infection [39,40,41].

According to the data available in the literature, the resistance of *H. pylori* to the classes of antibiotics used is different depending on the geographical area and the national health programs, giving results that tend to be uniform at the global level and an increase in antibiotic resistance for *H. pylori* regardless of local socio-economic status. Triple therapy regimens (with clarithromycin and amoxicillin) are most commonly prescribed. However, they have a failure rate estimated at over 20–30% of patients, a failure that appears due to an increasing resistance to clarithromycin.

Quadruple therapy regimens without bismuth (PPI, amoxicillin, clarithromycin, and metronidazole) have been recommended with improved eradication rates, with the mention that this regimen is not as effective in the case of double resistance of both metronidazole and clarithromycin. Available literature estimates that over the last 20 years, clarithromycin resistance has increased yearly by 3.7% [42].

Associating bismuth with therapeutic regimens is beneficial, especially in areas with increased antibiotic resistance. Bismuth has a solid bacteriostatic effect with prominent synergic benefits in various therapies when combined with several antibiotics. Bismuth is most associated with the quadruple therapy regimen, along with PPIs, tetracycline, and metronidazole. The association of bismuth with triple therapy regimens also improved *H. pylori* infection eradication rates [15].

The excessive use of macrolides in treating respiratory infections (azithromycin, erythromycin) and cross-resistance between antibiotics may be responsible for the increase in microbial resistance to clarithromycin observed in more and more geographical areas in cohort studies [3].

Consequently, various therapeutic regimens have been used to treat *H. pylori*. In geographical areas with high resistance to clarithromycin, the combination of metronidazole concurrently with PPIs, clarithromycin, and amoxicillin twice a day (a quadruple therapy regimen) increases the efficacy of the treatment, with an eradication rate of over 90% [43]. Clarithromycin can be replaced by levofloxacin (250/500 mg) in triple therapy regions, with clarithromycin resistance above 15–20% and quinolone resistance below 10%, therapy will increase the eradication rate [44,45]. The frequent, exhaustive, and sometimes incorrect use of antibiotics influenced the selection of multi-resistant germs. Macrolides and fluoroquinolones, used to eradicate *H. pylori,* are also widely used in other infectious pathologies; prescribing them should be performed judiciously. For example, the increasing prescription of quinolones in urinary tract infections has increased Escherichia coli resistance, leading to the emergence of multidrug-resistant species [46].

### 6.2. Probiotics

A new approach to *H. pylori* therapy is the association of probiotics, based on multiple clinical studies focused on an association that has become imperative, mainly due to the increase in antibiotic resistance. Probiotics are live microbial species with anti-inflammatory and antioxidant characteristics that can improve the intestinal microbiome, antagonize the action of pathogens, and modulate the immune status [47]. Frequently used probiotic bacteria are Lactobacillus and Bifidobacterium species [48].

According to the studies carried out by Kim et al., the prescription of probiotics for *H. pylori* infection increases the eradication and reduces the side effects of classical antimicrobial therapy. A common characteristic of these bacteria is their capacity of the anaerobic digestion of carbohydrates and the production of lactic acid. These microorganisms have an increased resistance to low gastric pH and a tolerance to wide temperature variations [49].

Previous studies focused on the significance of probiotics (*Lactobacillus*, *Saccharomyces boulardii*, or *Bacillus clausii*) in improving side effects associated with *H. pylori* eradication therapy [50,51,52,53,54,55,56,57,58].

The benefits of probiotic therapy in *H. pylori* infection are an increased eradication rate and improved treatment compliance by preventing associated side effects [55].

Several factors, such as the type of *H. pylori* strain, the extent of the inflammation, and the density of *H. pylori* colonization, determine the clinical impact of H. pylori infection. The risk of peptic ulcers and gastric cancer increases with the extent of the infection. For this reason, the effort to eradicate *H. pylori* will decrease the risk of diseases associated with *H. pylori* infection, justifying the association of probiotics with classical therapies [59,60].

### 6.3. Mechanism of Action of Probiotics

#### 6.3.1. Non-Immunological Mechanism

The acidic pH of the stomach is the first line of defense against pathogenic bacteria that penetrate the gastric mucosal barrier. The administration of probiotics amplifies this protection, thanks to the generation of antimicrobial substances that compete with *H. pylori* for adhesion receptors, increasing mucin production and stabilizing the gastric mucosa barrier [61] (Figure 4).

##### Antimicrobial Substances

Probiotics release short-chain fatty acids that inhibit *H. pylori* development. Releasing short-chain fatty acids produced during carbohydrate metabolism, such as acetic, propionic, and lactic acids, determines a reduction in gastric pH.

Antimicrobial compounds related to the bacteriocin classes are synthesized by certain Lactobacillus species (Figure 4). Bacteriocins possess a peptide structure with antimicrobial potential and are implicitly anti-*H. pylori* [62].

##### Competition for Adhesion

The mechanisms of the inhibition of the adhesion of *H. pylori* to the gastric epithelial cells by probiotics depend on the state of the epithelial mucosa, the density of receptors associated with the gastric mucosal adhesion, and the host’s immune status (Figure 4).

##### The Mucosal Barrier

Mucous surfaces have their own capacity to protect against noxious and pathogenic agents in the intestinal lumen. Mucins are formed by complex glycoproteins that protect the intestinal mucosa from microbial pathogens [63]. A study performed in 2014 by Hanish et al. described the ability of *H. pylori* to suppress mucin gene expression in gastric cells. This ability of mucins restores the permeability of the gastric mucosa membrane and inhibits the adhesion of *H. pylori* to the epithelial cells [64] (Figure 4).

#### 6.3.2. Immunological Mechanisms

The release of different inflammatory mediators, such as chemokines and cytokines, represents a characteristic inflammatory response to the infection caused by *H. pylori.* The role of probiotics consists of modulating the immunological response by amplifying the secretion of anti-inflammatory cytokines, leading to a reduction in gastric inflammation [65] (Figure 4).

Lactobacillus, Bifidobacterium, and Saccharomyces are beneficial probiotics in eradicating *H. pylori* and reducing the side effects of antimicrobial therapy, such as nausea, vomiting, diarrhea, and epigastric pain, improving drug tolerance and patient compliance. Probiotics should be administered concurrently with standard triple therapy due to the significant increase in the *H. pylori* eradication rate and the reduction in side effects.

### 6.4. Efficacity of Different Probiotics

The association of Lactobacillus acidophilus/Bifidobacterium animalis, Lactobacillus acidophilus/Bifidobacterium bifidum, Lactobacillus helveticus/Lactobacillus rhamnosus, and Lactobacillus acidophilus/Bifidobacterium longum/Enterococcus faecalis administered in association with standard triple therapy reduced the incidence of secondary effects and the occurrence of antibiotic-associated diarrhea. Emara et al. studied the role of probiotics in histological improvement, demonstrating a decrease in H. pylori density on the luminal part of the epithelium, thus improving the histological degree of inflammation in the corpus and gastric antrum [66].

On the other hand, the eradication of *H. pylori* could be improved by the intestinal microbiota, according to available data [16,67]. The association of probiotics with the triple therapy scheme (*Streptococcus faecium* and *Bacillus subtilis*) can reduce the occurrence of antibiotic-resistant bacterial strains [68].

Guillemard et al. conducted a randomized controlled trial on 136 adults undergoing 14 days of *H. pylori* triple therapy, using fermented milk containing yogurt strains and *Lactobacillus paracasei* CNCM I-1518, *Lactobacillus paracasei* CNCM I-3689, and *Lactobacillus rhamnosus* CNCM I-3690 as a probiotic. They suggested the probiotic had a protective effect, preserving short-chain fatty acid production and gut microbiota homeostasis. However, the product’s effect on antibiotic-associated diarrhea and gastrointestinal symptoms was negligible, as both occurrences had an unexpectedly low frequency in the control population [69].

A prospective randomized placebo-controlled trial conducted by Viazis et al. evaluated the efficacy of the combined use of four probiotic strains—*Lactobacillus acidophilus*, *Lactiplantibacillus planatarum*, *Saccharomyces boulardii*, and *Bifidobacterium lactis*. The probiotic mixture was administered to patients receiving a quadruple eradication regimen. They discovered that probiotic use yielded an increase in *H. pylori* eradication rate (92.0% in the test group vs. 86.8% in the control group) and a significant decrease in treatment-associated severe symptoms (1.2% in the test group vs. 14.6% in the control group) [70].

Chen et al. measured the effect of probiotics on the bacterial load of *H. pylori* in a double-blinded, randomized, controlled trial. The researchers used only strains of *Lactobacillus acidophilus* and *Lactobacillus rhamnosus*, without PPIs or antibiotics. They concluded that probiotics lowered the bacterial load for the duration of the treatment in the test group, but with no significant difference between groups 2 weeks after the treatment stopped. Eradication of *H. pylori* was observed in none of the patients [71].

## 7. Particular Aspects of *H. pylori* Infection in Pediatric Patients

In industrialized countries, the incidence of infection in the pediatric field increases with age [72]. According to epidemiological studies, *H. pylori* infection in children is influenced by environmental factors, factors of the gastric mucosa of the host, and bacterial virulence [73], and it could be carried throughout life if left untreated.

According to the guidelines of the European Society of Pediatric Gastroenterology and Nutrition (ESPGHAN)/North American Society of Pediatric Gastroenterology and Nutrition (NASPGHAN), a positive culture accompanied by another examination (a rapid urease test) is necessary for diagnosis.

The risk factors related to the infection are represented by environmental and family characteristics, such as hygiene factors, sanitary conditions at home, the number of people living in the house, and the family history of dyspeptic disorders. Other risk factors are represented by the history of previously prescribed antibiotic treatments for other infectious pathologies (e.g., respiratory tract, which is common in children). The preferred administration of clarithromycin or amoxicillin/clavulanic acid determined the increase in antibiotic resistance of *H. pylori.* Recent studies propose dental plaque and caries as reservoirs for *H. pylori* in the pediatric population [74,75].

The infection appears around ten years of age, with most patients being asymptomatic for long periods. Clinical manifestations are non-specific, and some may be justified only by the presence of complications. Frequent symptoms such as nausea or epigastric pain may decrease in frequency and intensity or even disappear with or without the eradication of the bacteria. A meta-analysis documented a statistically significant association with epigastric pain but no statistically significant association with other nonspecific digestive symptoms, such as vomiting, diarrhea, flatulence, abdominal pain, or constipation [76].

In a recent study, Kolasa-Kicińska et al. highlighted the association between *H. pylori* infection and idiopathic short stature. They detected lower insulin-like growth factor 1 and ghrelin levels in infected children, hypothesizing that the decline in growth rate in this subpopulation is directly caused by diminished concentrations of these two hormones [77].

The literature supports the link between iron deficiency anemia and *H. pylori* infection in pediatric patients. However, the direct connection between these two pathological entities is not clear, and causality has not yet been proven. Likely, this association is simply due to the low socio-economic status of the analyzed cohorts, predisposing patients to both iron deficiency and *H. pylori* infection [78].

An association between chronic tonsillitis and *H. pylori* infection was recently described in the literature. This association is not present in the adult population, but only in children. Researchers state that chronic tonsillitis could be relevant to pediatric *H. pylori* infection [79]. Another association observed is between *H. pylori* and celiac disease. A recent study confirmed this correlation but could not clarify the causality, emphasizing the need for future research [80].

One cross-sectional study by Wang et al. suggests a negative correlation between asthma and *H. pylori* in pediatric patients. In the studied cohort, 3.77% of the *H. pylori*-positive group were diagnosed with asthma, while it was diagnosed in 7.23% of the *H. pylori*-negative group [81].

Symptoms such as vomiting, a digestive hemorrhage, iron-deficiency anemia, and malnutrition may be due to complications of *H. pylori* infection or other etiological diagnoses and therefore require further investigation.

The positive diagnosis in children must be performed using invasive methods, and the response evaluation after the eradication therapy using non-invasive methods [82].

The most common endoscopic diagnosis in children is nodular gastritis. Initially, *H. pylori* colonizes the antrum and can produce antral gastritis. Unlike in adults, peptic ulcers in children are rare. Gastric atrophy and intestinal metaplasia are less common in children than in adults [83].

The sensitivity of all invasive methods is influenced by antibiotics, bismuth, proton pump inhibitors (PPIs), and upper gastrointestinal bleeding. Culture has a high specificity of 100%, but the sensitivity is diminished by antibiotics, PPIs, bismuth, and active digestive bleeding, and a negative culture does not exclude the diagnosis [19,84]. PCR offers excellent specificity and sensitivity, allowing the detection of specific mutations that lead to antibiotic resistance. RUT is a rapid method with high sensitivity and specificity, even though the sensitivity decreases in young children after using antibiotics, PPIs, and bismuth. Histology allows a positive diagnosis and appreciates the degree of chronic inflammation, lymphoid follicles, atrophic gastritis, and intestinal metaplasia. Neutrophil infiltration is less significant in children than in adults, but infiltration by lymphocytes, plasma cells, and immunosuppressive regulatory T cells is predominant [19,84].

Non-invasive tests are not the first step in detecting *H. pylori* infection in children, except in cases where endoscopy cannot be performed. The stool antigen test and the urea breath test are used to evaluate eradication and are indicated four weeks after therapy. If SAT is positive after treatment, it is recommended to perform UBT in children. Some studies suggest a non-invasive test evaluation annually, especially in high-prevalence areas or in patients at risk [85].

*H. pylori* serological tests in serum, blood, urine, and saliva are not modified by PPI or antimicrobial treatment and have a high sensitivity. Still, they are used for epidemiological purposes with anti-*H. pylori* antibodies remaining positive for a while after eradication, so they do not allow for differentiation between current and chronic infection.

The treatment of *H. pylori* infection in pediatric patients is based on protocols of the ESPGHAN/NASPGHAN guidelines, revised in 2016.

The first-line treatment is recommended according to the antimicrobial susceptibility of the *H. pylori* strain or the tendency of antimicrobial resistance in the respective region. Eradication control should preferably be performed through a non-invasive test (a urea breath test in children over six years old or a stool antigen test) 4–8 weeks after the end of the antibiotic treatment. The main causes of treatment failure are antimicrobial resistance, side effects of therapy, and low compliance to the treatment in the pediatric population [82,86,87]. One novel finding suggests that the expression of inflammatory cytokines heavily influences the success rate of *H. pylori* eradication therapy. Raised concentrations of IL-1β, IL-6, and TNF-α seem to represent risk factors for the failure of eradication [88].

Without a treatment regimen, eradication is highly unlikely. Spontaneous eradication is mainly described in young children, but the eradication rate reduces with age [89].

Standard triple therapy based on clarithromycin is widely prescribed, especially in the primary care network. Alternative regimens are only recommended in specialist centers. The widespread use of antibiotics (clarithromycin for respiratory infections) in the general population has increased antibiotic resistance [82,86,87,90,91].

Future studies evaluating compliance with different treatment regimens, higher PPI doses, and quadruple therapies are sometimes needed, particularly in patients with unknown susceptibility, multidrug-resistant strains, or triple therapy failure.

Available data on probiotics in *H. pylori* infection suggest that Lactobacillus supplementation to therapy at high doses and for long periods decreases the risk of adverse effects, such as diarrhea [92].

The administration of probiotics as supportive therapy along with the standard regimen reduced the eradication rate by approximately 13% (84.0% versus 71.4%). According to a meta-analysis by Fang et al. [92], it may reduce the incidence of therapy-associated diarrhea.

Efforts to investigate and treat *H. pylori* infection in children should be based on providing tangible benefits, such as lower complication rates, interrelation with allergic, parasitic, or immunological diseases, antimicrobial resistance, and the few treatment alternatives available to the pediatric population.

## 8. Conclusions

The prevalence of antibiotic resistance to clarithromycin and metronidazole in *H. pylori* infection is not negligible, a finding that highlights the need for a new orientation of gastroenterology teams in diagnosing and treating *H. pylori* infections.

The exhaustive treatment with antibiotics increases the pressure on the medical community, so implementing local and national screening and surveillance programs and developing new non-invasive techniques in clinical practice are imperative to amplify the efficacy of chronic *H. pylori* infection treatment.

*H. pylori* infection remains a vital pathogen implicated in digestive pathology. A better understanding of the behavior of the immune system at different ages, favoring the persistence of infection in childhood, and then gastric complications in adults, may help to develop global strategies for reducing the prevalence of *H. pylori* to prevent the long-term adverse effects associated with *H. pylori* infection.

Associating probiotics with traditional treatment regimens increases the eradication rate due to the reduced side effects of antibiotics and improved treatment compliance. However, this effect might be specific to the strain, dose, or duration of the therapy.

A fascinating perspective hypothesis is the administration of probiotics to reduce *H. pylori* adhesion to gastric epithelial cells, preventing *H. pylori* colonization, especially in children, or reinfection with *H. pylori* in high-risk patients.

Researchers’ results are encouraging, but further clinical trials are needed. Additional studies will clarify the appropriate probiotic strain choice, dosage, and duration of administration.

## Figures and Tables

**Figure 1 antibiotics-12-00060-f001:**
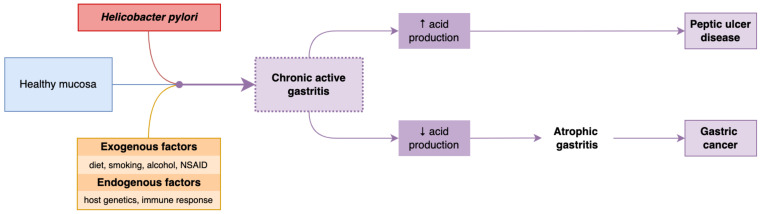
Physiopathology of *H. pylori* infection.

**Figure 2 antibiotics-12-00060-f002:**
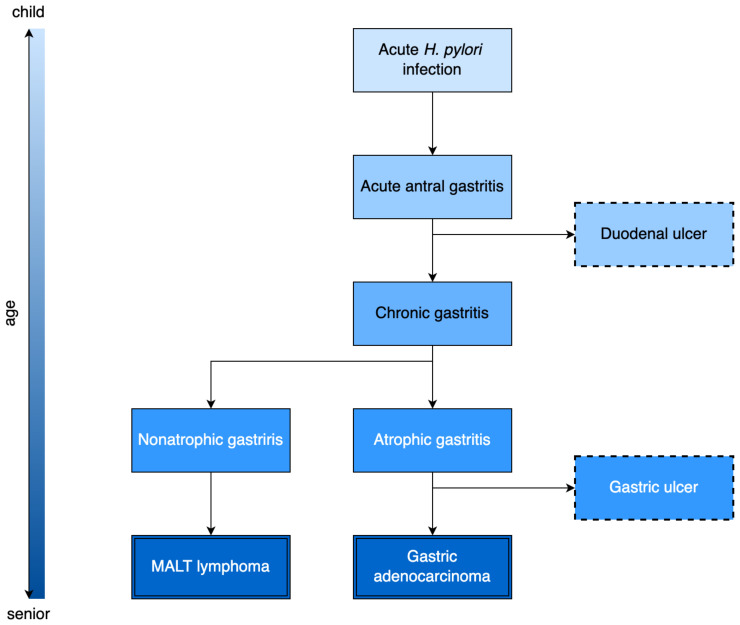
Timeline of clinical manifestations in patients infected with *H. pylori*.

**Figure 3 antibiotics-12-00060-f003:**
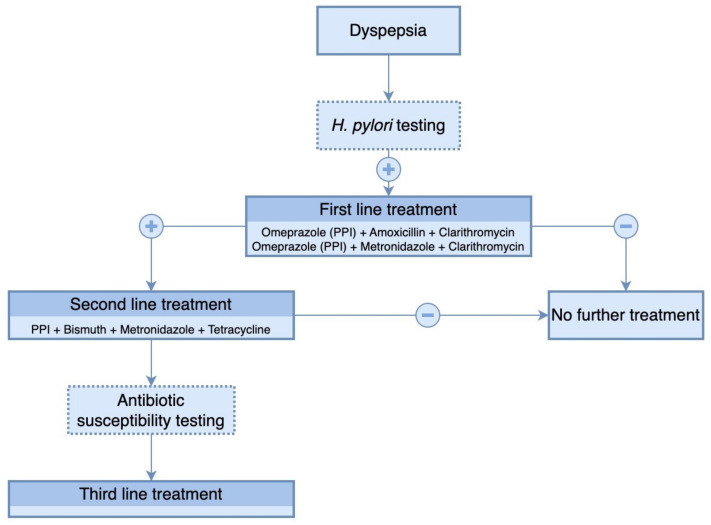
Therapeutic algorithm for eradication of *H. pylori* infection.

**Figure 4 antibiotics-12-00060-f004:**
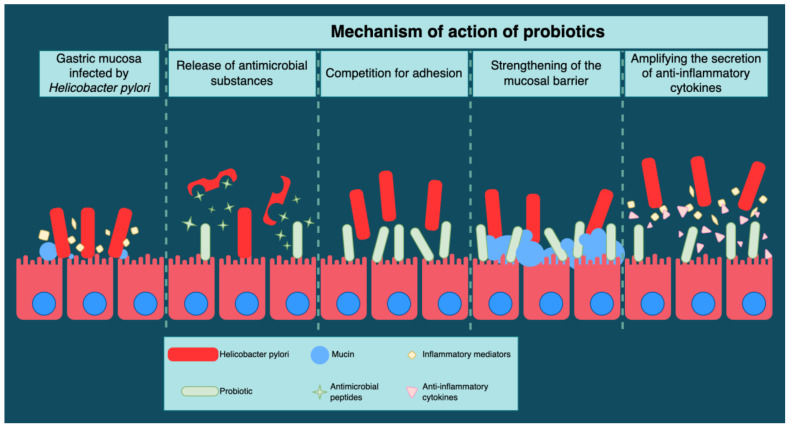
Probiotics’ mechanism of action in *H. pylori* therapy.

## Data Availability

Not applicable.

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
