# Peer review of "The Importance of Accurate Early Diagnosis and Eradication in *Helicobacter pylori* Infection: Pictorial Summary Review in Children and Adults"

_antibiotics, 2022, doi:10.3390/antibiotics12010060_

Round 1

Reviewer 1 Report

The articlre emphasizes the importance of accurate early diagnosis and eradication in H pylori infection:, but the information is not new and it is already being applied in practice. They are international guidelines that help practitioners.
The introduction is too short.
The work methodology is not specified in your article.
H. pylori should be written in italics.
The same abbreviation should be maintained, HP or H. pylori.
Anyway, there are technical editing mistakes.

Author Response

Thank you for taking the time to review this manuscript.At your suggestion, we have added more information in the introduction, and as well we have specified the methodology. H.pylori is now written in italics.

We hope our manuscript's new version is much better than the previous one. Thank you so much for these constructive indications.

Kind regards,

Corina Vasile

MD, PhD

Reviewer 2 Report

The review turns out to be well written, and it is characterized by a description of the literature accurately and punctually, although it is missing in the same sections of important concepts and references (i.e. in "Prevalence and mode of transmission" and "Pathogenesis" paragraphs). Unfortunately, the publication is repetitive and redundant compared to many other reviews already in the literature about H pylori infection. This publication is not distinguished by details.

Author Response

Dear reviewer,

Thank you for the time you took to review our manuscript and for the constructive suggestions. Our manuscript is the only one that is presenting information about both children and adults. We have made some adjustments following your indications.

Kind regards,

Corina Vasile

MD, PhD

Reviewer 3 Report

Review report:

TITLE: The importance of accurate early diagnosis and eradication in H pylori infection: pictorial summary review in children and adults

Journal: Antibiotics

To editorial office

Here I am sending you my comments on recently submitted paper by Marginean et al to the journal “antibiotics” entitled the importance of accurate early diagnosis and eradication in H pylori infection: pictorial summary review in children and adults”. Having the first overview on the paper I got the impression that the review is written nicely and it can be even better after some rounds of revision. My ideas and suggestions are designed to help them all for having a better shaped paper accepted in this journal.

1-      A minor polish for English edits seems helpful to avoid some tiny mistakes but they are not too much.

2-      Abstract: using the HP abbreviate means you should not add the bacterial full name in the next line! Please correct it.

3-      A vague sentence for readers, please rephrase it “A fascinating perspective hypothesis is an administration of probiotics to reduce H. pylori adhesion to gastric epithelial cells, preventing H. pylori colonization, especially in children, or reinfection with H. pylori in high-risk adult patients.”

4-      Keywords: the last key word is not necessary, remove it

5-      Introduction: sentence “In 1983, H. pylori was isolated from the gastric antrum in 1983, by Marshall and Warren being described as a gram-negative flagellate and microaerophilic bacterium” used the 1983 twice while one of them is not needed, correct it in revised version.

6-      In the section of probiotics, please add some sentence about newest findings about probiotics studies supporting the new combination of the probiotics in therapies against this bacterium in infected cases.

7-      Page 10: section “7. Particular aspects of H. pylori infection in pediatric patients” I recommend to talk more about clinical aspects of infection with hp in this population. I think the authors aimed to discuss it but nothing new  was found. Sorry!

8-      Tables and figures are totally fine by me.

9-      Page 2, line 73: the sentence “Studies published so far estimate that approximately 10% of subjects infected with H. pylori develop peptic ulcers, and 1%-3% of subjects develop gastric cancer” needs referencing, I provided you some suggestions, please amend it.

* Backert, S. and Clyne, M., 2011. Pathogenesis of Helicobacter pylori infection. Helicobacter16, pp.19-25.

** Abadi, A.T.B., Ierardi, E. and Lee, Y.Y., 2015. Why do we still have Helicobacter pylori in our stomachs. The Malaysian journal of medical sciences: MJMS22(5), p.70.

10-  Page 8, line 316: talking about clarithromycin as one of members of anti-H. pylori therapy needs further cautious since it got higher rate of resistance after 2017. There are more reports confirming this message which should be somehow reflected in this paper, I suggest, se ethe below;

* Schubert, J.P., Warner, M.S., Rayner, C.K., RobertsThomson, I.C., Mangoni, A.A., Costello, S. and Bryant, R.V., 2022. Increasing Helicobacter pylori clarithromycin resistance in Australia over 20 years. Internal Medicine Journal52(9), pp.1554-1560.

11-  Page 12. “A better understanding of the behavior of the immune system at different ages, favoring, in childhood, the persistence of infection and then, in adults, gastric complications, may help to develop global strategies for reducing the prevalence of H. pylori, such as vaccines, and to prevent long-term adverse effects associated with HP infection.” I am not agree with recommendation of vaccine to efficiently control the increased rate of gastric cancer worldwide. This suggestion is lonely considerable in region with high rate of resistance and report of gastric cancer like Japan. So such conclusion should be amended in new version, also stated in slighter language.

Author Response

Dear reviewer,

Thank you for taking you time to review our work and especially for the suggestions you have given.

  • A minor polish for English edits seems helpful to avoid some tiny mistakes but they are not too much. 

We have performed an English revision.

  • Abstract: using the HP abbreviate means you should not add the bacterial full name in the next line! Please correct it.

We have corrected this typo. Thank you.

  • A vague sentence for readers, please rephrase it “A fascinating perspective hypothesis is an administration of probiotics to reduce H. pylori adhesion to gastric epithelial cells, preventing H. pylori colonization, especially in children, or reinfection with H. pylori in high-risk adult patients.”

We have rewritten this sentence, thank you.

“Apparently, the administration of probiotics in children can reduce H. pylori adhesion to epithelial cells preventing colonization, and in adult patients, can prevent reinfection. “

  • Keywords: the last key word is not necessary, remove it

We have removed it. Thank you.

  • Introduction: sentence “In 1983, H. pylori was isolated from the gastric antrum in 1983, by Marshall and Warren being described as a gram-negative flagellate and microaerophilic bacterium” used the 1983 twice while one of them is not needed, correct it in revised version.

We revised the introduction sentence.

“In 1983, H. pylori was isolated from the gastric antrum by Marshall and Warren being described as a gram-negative flagellated and microaerophilic bacterium.”

  • In the section of probiotics, please add some sentence about newest findings about probiotics studies supporting the new combination of the probiotics in therapies against this bacterium in infected cases.

We added more data in the Probiotics section. Thank you for this suggestion.

  • Page 10: section “ Particular aspects of H. pylori infection in pediatric patients” I recommend to talk more about clinical aspects of infection with hp in this population. I think the authors aimed to discuss it but nothing new  was found. Sorry!

We extended this section as well.

  • Tables and figures are totally fine by me.

Thank you !

9-      Page 2, line 73: the sentence “Studies published so far estimate that approximately 10% of subjects infected with H. pylori develop peptic ulcers, and 1%-3% of subjects develop gastric cancer” needs referencing, I provided you some suggestions, please amend it.

* Backert, S. and Clyne, M., 2011. Pathogenesis of Helicobacter pylori infection. Helicobacter16, pp.19-25.

** Abadi, A.T.B., Ierardi, E. and Lee, Y.Y., 2015. Why do we still have Helicobacter pylori in our stomachs. The Malaysian journal of medical sciences: MJMS22(5), p.70.

      We extended our references with  the suggested ones. Thank you.

10-  Page 8, line 316: talking about clarithromycin as one of members of anti-H. pylori therapy needs further cautious since it got higher rate of resistance after 2017. There are more reports confirming this message which should be somehow reflected in this paper, I suggest, se ethe below;

* Schubert, J.P., Warner, M.S., Rayner, C.K., Roberts‐Thomson, I.C., Mangoni, A.A., Costello, S. and Bryant, R.V., 2022. Increasing Helicobacter pylori clarithromycin resistance in Australia over 20 years. Internal Medicine Journal52(9), pp.1554-1560.

We revised this paragraph.

11-  Page 12. “A better understanding of the behavior of the immune system at different ages, favoring, in childhood, the persistence of infection and then, in adults, gastric complications, may help to develop global strategies for reducing the prevalence of H. pylori, such as vaccines, and to prevent long-term adverse effects associated with HP infection.” I am not agree with recommendation of vaccine to efficiently control the increased rate of gastric cancer worldwide. This suggestion is lonely considerable in region with high rate of resistance and report of gastric cancer like Japan. So such conclusion should be amended in new version, also stated in slighter language.

      We have revised this part as well. Thank you.

Kind regards,

Corina Vasile

MD, PhD

Round 2

Reviewer 1 Report

Still a lot of missing italics for Helicobacter pylori and H. Pylori even in the title... the paper should be revised by a native speaker or editing service...

Urea rapid test is not expansive...

13 authors seem too much for a 14 page review, you should limit to 6 or 8...

Still nothing about the methodology, what databases you searched, what search keywords you used, how you selected the articles you cited...

Author Response

Dear reviewer,

We apologize for not being extraordinarily vigorous and letting some H. pylori without italics. We hope we have corrected all of them.

The urea rapid test is not expansive. You are right, we modified it.

Yes, 13 authors for 14 page review it’s pretty much a, but since all of the authors worked for this manuscript, it’s not fair to remove some of them …. We will keep that in mind for the following papers.

Since it’s a narrative review, there is not much to be described…

Thank you,

Corina Vasile

MD, PhD

Clinical and Research Fellow at Pediatric and Adult Congenital Department, University Hospital of Bordeaux

Reviewer 2 Report

None

Author Response

Thank you